# Mechanical Behavior of Ti6Al4V Scaffolds Filled with CaSiO₃ for Implant Applications

**Ramin Rahmani [1],\*, Maksim Antonov [1], Lauri Kollo [1], Yaroslav Holovenko [1] and Konda Gokuldoss Prashanth [1,2,3],\***

[1] Department of Mechanical and Industrial Engineering, Tallinn University of Technology, Ehitajate tee 5, 19086 Tallinn, Estonia; maksim.antonov@taltech.ee (M.A.); lauri.kollo@taltech.ee (L.K.); yaholo@taltech.ee (Y.H.)

[2] Erich Schmid Institute of Materials Science, Austrian Academy of Science, Jahn Straße 12, A-8700 Leoben, Austria

[3] CBCMT, School of Mechanical Engineering, VIT University, Vellore, Tamil Nadu 632014, India

\* Correspondence: ramin.rahmaniahranjani@taltech.ee (R.R.); kgprashanth@gmail.com (K.G.P.)

**Abstract:** Triply periodic minimal surfaces (TPMS) are becoming increasingly attractive due to their biomedical applications and ease of production using additive manufacturing techniques. In the present paper, the architecture of porous scaffolds was utilized to seek for the optimized cellular structure subjected to compression loading. The deformation and stress distribution of five lightweight scaffolds, namely: Rectangular, primitive, lattice, gyroid and honeycomb Ti6Al4V structures were studied. Comparison of finite element simulations and experimental compressive test results was performed to illustrate the failure mechanism of these scaffolds. The experimental compressive results corroborate reasonably with the finite element analyses. Results of this study can be used for bone implants, biomaterial scaffolds and antibacterial applications, produced from the Ti6Al4V scaffold built by a selective laser melting (SLM) method. In addition, Ti6Al4V manufactured metallic lattice was filled by wollastonite (CaSiO₃) through spark plasma sintering (SPS) to illustrate the method for the production of a metallic-ceramic composite suitable for bone tissue engineering.

**Keywords:** Ti6Al4V scaffolds; triply periodic minimal surfaces; selective laser melting; additive manufacturing; biomaterial applications; finite element analysis; spark plasma sintering; wollastonite

## 1. Introduction

Titanium and its alloys have been widely used for biomedical and orthopedic applications, such as hip, knee, femur, vertebra, bone and skull, due to their excellent antibacterial/biocompatibility, strength-to-weight ratio and wear and corrosion resistance in comparison with stainless steel [1–5]. Implants have been made by computer numeric control (CNC) machining, or powder metallurgy, followed by different post-processing procedures. The metallic biomaterial scaffolds are applied for stent placement [6] or bone replacement [7]. Three dimensional (3D) printed porous scaffolds are suitable for seeding cells, delivering drugs, and are able to carry compression/tensile loading [2,3]. Biological activities along the scaffold surface (cell growth) and structural design of the scaffolds for additive manufacturing (AM) are both very important, and extensive research has been conducted [8]. With the use of optimal scaffold material and the suitable design for AM, scaffolds with added functionalities, elastic modulus, and strength matching the human bone can be realized [9].

Triply periodic minimal surfaces (TPMS) are some of the designs that are expected to be optimal candidates for bone tissue engineering applications. Primary simulation of the TPMS are crucial for designing these scaffolds to figure out the stress distribution, deformation and failure mechanism of porous structure [10,11].

Gradient structures, unit cell size, strut diameter, the volume fraction the of sample, plasticity and damage tolerance, as well as the minimizing of cost/time are considered during the computer-aided design (CAD) and simulation. In order to have optimized lightweight structures under loading and in vivo or in vitro conditions, printed scaffolds can be graded axially or radially [12]. Recently, research is focused upon polymeric printing using polyjet/inkjet deposition [13], which is faster with no post-processing, unlike metal printing. However, the low mechanical properties of polymer 3D printed materials make it non-applicable for in vivo conditions. Porous Ti6Al4V structures have yield strength and compressive strength in the range of 90–220 MPa [14]. The computer-aided design (CAD) of porous scaffolds and finite element analysis (FEA) have recently attracted much attention in the tissue engineering field. Fluid permeability [15] and biocompatibility [16] of scaffolds depends directly on the pore geometry and topology [17] of structures. The curved shape, smoothness, good flowability, 3D manufacturability and biocompatibility of Ti6Al4V powder particles makes Ti-based alloy promising for load bearing bone implants [18]. It is well known that the artificial bone scaffolds should have the following characteristics: (1) Biocompatibility with living tissues; (2) mechanical properties for trabecular bone and cortical bone (ranging from 0.7 to 15 MPa till >100 MPa [19]); and (3) porous fabricated structures for osteoporosis diseases. Periodic porous materials like TPMS are potential candidates for biomimetic scaffold architecture [19]. The main advantage of TPMS scaffolds is the open interconnected cell structure, deemed to facilitate cell migration, vitalization and vascularization, while retaining a high degree of structural stiffness [20]. Henceforth, mechanical properties, especially compressive strength, is considered as an important parameter for TPMS scaffolds. In this paper, we have performed compressive tests, where the experimental results are compared with the FEA for the selected promising TPMS structures.

Ti6Al4V is considered as excellent biomaterial because of higher fatigue strength, tensile strength and compressive strength [21–24]. However, it is bioinert, where an improper integration takes place with the host bone, resulting in a weak interfacial bond. As a result, at the bone-implant interface, there can be an accumulation of the necrotic fibrous tissue [25]. Nevertheless, in vivo life of the load bearing scaffolds can be increased by improving the interfacial bond of the bone and the metallic scaffolds. To improve the interfacial bonding, Ti6Al4V scaffolds are filled with wollastonite. Spark plasma sintering (SPS) was used for integrating wollastonite into the cellular Ti6Al4V structure [26]. Highly porous acicular wollastonite ($CaSiO_3$) with micro- or nano-sized pores results in high corrosion resistance and good biocompatibility [27,28].

Hence, the present study aims to fabricate Ti6Al4V scaffolds reinforced with bioactive elements (wollastonite) for the proper integration of the implant to the bone via tissue growth. By reviewing the articles, the deficiency of comparison between the mechanical properties of scaffolds, FEA, and an assortment of them regarding the applications is tangible. Five different Ti6Al4V scaffolds (samples) with identical outside dimensions and similar weight, namely: Rectangular, primitive, lattice, gyroid and honeycomb, were considered. They were chosen based on the desire to provide a continuous surface that is important for the growth/cultivation of viable cells and other aspects of tissue engineering [29]. The rectangular type has vertical available surface, whereas the primitive is made by sequential hollow spheres (proper for fluid permeability and drag delivery). Therefore, scaffolds A and B have vertical flat and spherical areas to flourish the cells. In the case of the lattice type structure, it is possible to specify the size of the cell and the diameter of the strut to figure out the proper cylindrical surface of the metallic rods. It is an intersecting cellular structure that is easy to use during the following of the SPS process, since it can be easily modeled, printed and filled by binder powders, and also allows a larger shrinkage required to produce hard metal-ceramic composites with a low porosity level by SPS [30]. The gyroid-type structure represents irregular continuous curves, and honeycomb considers embedded tubes, which are connected by horizontal washers.

A SolidWorks design and Ansys FEA combination were used to generate these structures and to further anticipate the mechanical behavior of the scaffolds subjected to compressive loading. The results from simulations are compared with the experimental compressive results.

## 2. Materials and Methods

Five distinct scaffolds were designed by SolidWorks, as shown in Figure 1 (rectangular, primitive, lattice, gyroid and honeycomb, respectively). The names are chosen according to the shape of the structures. These computer-aided design (CAD) models can be used both in finite element analysis (FEA) for the simulation of mechanical properties and the selective laser melting (SLM) process for the fabrication of the real parts. All the five scaffolds were of the following dimensions: A diameter of 20 mm and a height of 15 mm. The weight range of these samples varied between ≈3.8 and 5.5 g. Ti6Al4V powders with size of 10–45 μm (produced by a gas atomization process) from TLS Technik were used as the raw material for the production of triply periodic minimal surfaces (TPMS)-type scaffolds.

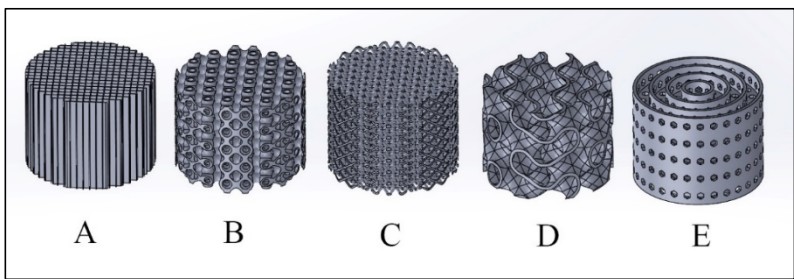

**Figure 1.** Scaffolds with: (**A**) Rectangular-, (**B**) Primitive-, (**C**) Lattice-, (**D**) Gyroid- and (**E**) Honeycomb-type structures (Dimensions: 20 mm in diameter and with 15 mm height), designed via computer-aided design (CAD) software.

A Realized SLM50 device with a maximum laser power of 120 W was used for the fabrication of the scaffolds. The following parameters were used for the fabrication of Ti6Al4V by SLM: Laser current 3000 mA, exposure time 600 μs, point distance 1 μm and unit cell size 1.5 mm. These are standard parameters that are observed for the fabrication of the Ti6Al4V samples. SLM-SPS processes were combined for producing metal-ceramic hybrid composites, which is suggested for chronic osteomyelitis, Vancomycin local delivery, and infected bones replacement in the field of tissue engineering [9]. The entire spark plasma sintering (SPS) setup is kept under the glovebox, in order to maintain the consolidation process in an inter atmosphere. This also helps in avoiding oxygen contamination during the SPS consolidation process. Microstructure of the samples were observed using scanning electron microscope (SEM), (Zeiss EVO MA 15, Germany)

Structural explicit dynamics using the AUTODYN solver and arbitrary Lagrange-Euler (ALE) method via ANSYS Workbench 17.2 is applied for these simulations. A high-quality finite element analysis (FEA) mesh is generated to gain a high-resolution response. The linearized governing motion equation in cylindrical coordinates ($r$, $\theta$, $z$) can be expressed by [31]:

$$\frac{\partial \sigma_{rr}}{\partial r} + \frac{1}{r}\frac{\partial \sigma_{r\theta}}{\partial \theta} + \frac{\partial \sigma_{rz}}{\partial z} + \frac{1}{r}(\sigma_{rr} - \sigma_{\theta\theta}) + F_r = \rho\frac{\partial^2 u_r}{\partial t^2} \tag{1}$$

$$\frac{\partial \sigma_{r\theta}}{\partial r} + \frac{1}{r}\frac{\partial \sigma_{\theta\theta}}{\partial \theta} + \frac{\partial \sigma_{\theta z}}{\partial z} + \frac{2}{r}\sigma_{r\theta} + F_\theta = \rho\frac{\partial^2 u_\theta}{\partial t^2} \tag{2}$$

$$\frac{\partial \sigma_{rz}}{\partial r} + \frac{1}{r}\frac{\partial \sigma_{\theta z}}{\partial \theta} + \frac{\partial \sigma_{zz}}{\partial z} + \frac{2}{r}\sigma_{rz} + F_z = \rho\frac{\partial^2 u_z}{\partial t^2} \tag{3}$$

where, $\sigma_{ij} = \sigma_{ji}$ is the Cauchy stress tensor, $u_i$ is displacement, $\rho$ is density and $F_i$ is body force ($F_r = F_\theta = 0$, $F_z = axial\ compression$). In the simulation, compressive punches are assumed rigid (unchanged dimension). Bottom punch is the fixed support and the top punch was kept moving until a maximum force of 100 kN (safe loading of machine), or a maximum deformation of 10 mm (with rate of 2 mm/min), is reached. Both the experimental tests and the numerical simulations have identical

boundary conditions. An Instron 8500 apparatus was used for measuring the compression tests of these scaffolds.

## 3. Biomaterial Production and Characterization

Figure 2 shows the five different TPMS scaffolds with a diameter of 20 mm and a 15 mm height. The five different TPMS scaffold types are: (a) Rectangular with a vertical flat area, (b) primitive, (c) lattice, (which is a very common type of structure produced using SLM [32]), (d) gyroid and (e) honeycomb (which is another common structure that is produced by SLM). Similarly Figures 3–7 exhibits the stress distribution and deformation in the five different TPMS scaffolds.

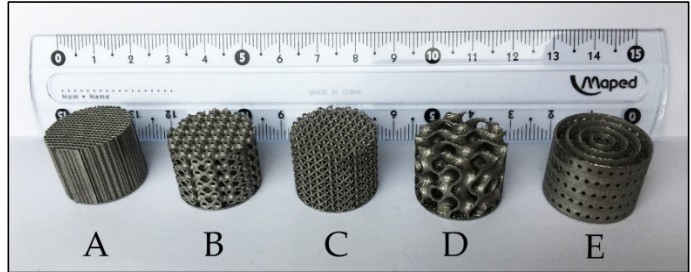

**Figure 2.** Scaffolds (their weight indicated in parentheses) with: (**A**) Rectangular (3.97 g), (**B**) Primitive (5.35 g), (**C**) Lattice (3.78 g), (**D**) Gyroid (4.04 g) and (**E**) Honeycomb (5.52 g) structures with 20 mm diameter and 15 mm height manufactured by selective laser melting.

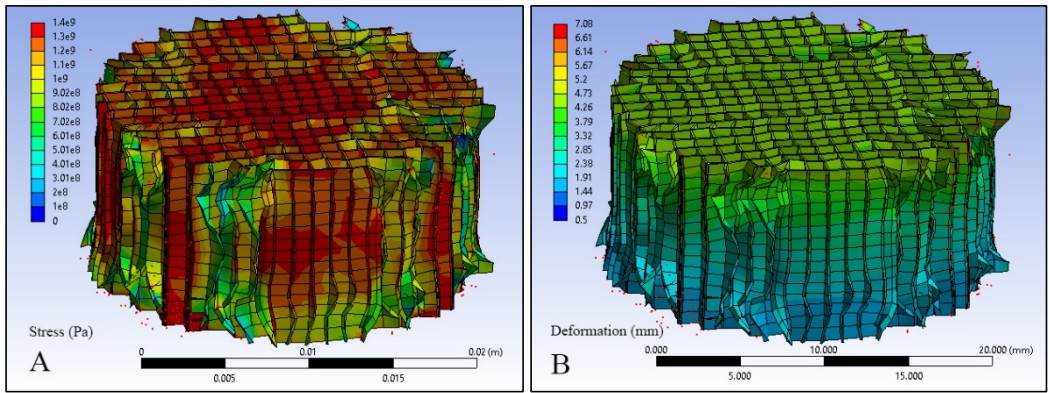

**Figure 3.** Rectangular scaffold simulation: (**A**) Stress and (**B**) displacement (final height after compression: 15 − 4.26 = 10.74 mm; red dots are separated particles in contact with compressive punches).

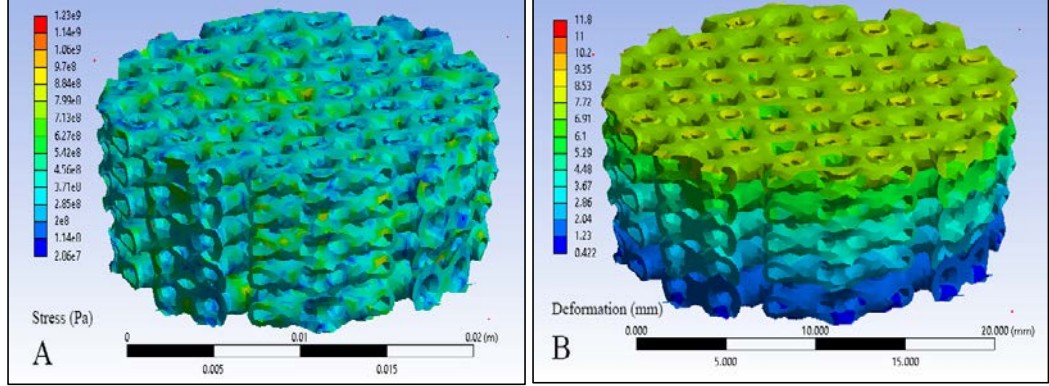

**Figure 4.** Primitive scaffold simulation: (**A**) Stress and (**B**) displacement (final height after compression 15 − 6.91 = 8.09 mm; the red dots are separated particles in contact with compressive punches).

The 99.9% purity wollastonite ($CaSiO_3$) with a particle size of 1–5 μm (supplied by NYAD, grade 1250) is filled and sintered (frittage) inside an argon-atomized Ti6Al4V cellular lattice structure via an SPS machine (made by FCT Systeme), as shown in Figure 8. The process was performed with an optimal pressure of 30 MPa at a temperature of 1100 °C, with a heating rate of 100 °C/min and a holding time of 5 min [33].

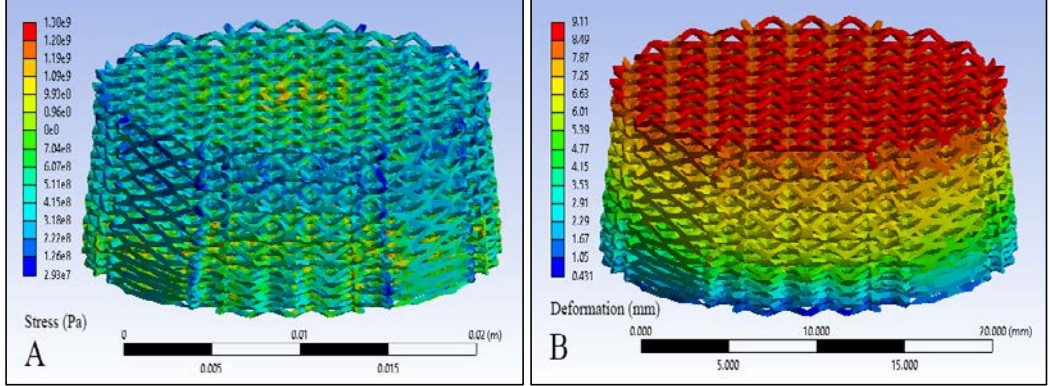

**Figure 5.** Lattice scaffold simulation: (**A**) Stress and (**B**) displacement (final height after compression 15 − 9.11 = 5.89 mm; red dots are separated particles in contact with compressive punches).

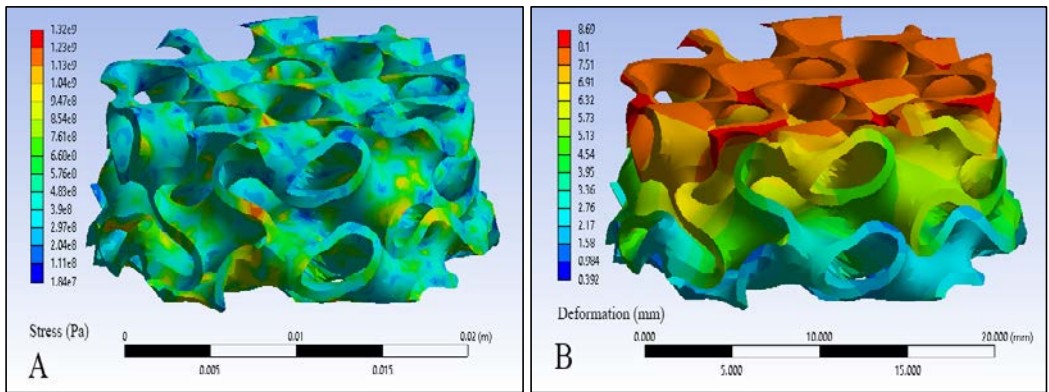

**Figure 6.** Gyroid scaffold simulation: (**A**) Stress and (**B**) displacement (final height after compression 15 − 8.1 = 6.9 mm; these red dots are separated particles in contact with compressive punches).

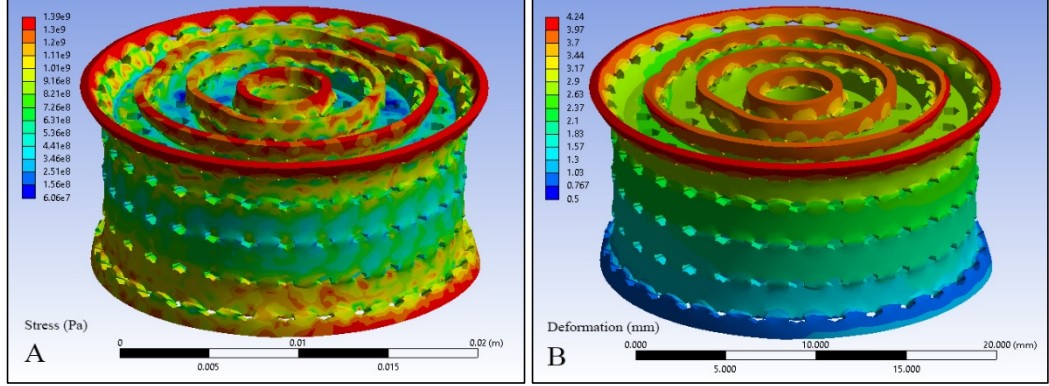

**Figure 7.** Honeycomb scaffold simulation: (**A**) Stress and (**B**) displacement (final height after compression 15 − 4.24 = 10.76 mm; red dots are separated particles in contact with compressive punches).

Wollastonite composition consists theoretically of 48.28% CaO and 51.72% SiO$_2$, while usually, the natural mineral may contain small amounts of iron, aluminum, magnesium, potassium and sodium (Figures 8A and 9A). An SLM-manufactured Ti6Al4V lattice with 1 mm cell size filled by CaSiO$_3$ powder and sintered in SPS (Figures 8B and 9B) are shown. The results show a good boundary between ceramic and lattice after SPS and polishing (Figures 8C and 9C).

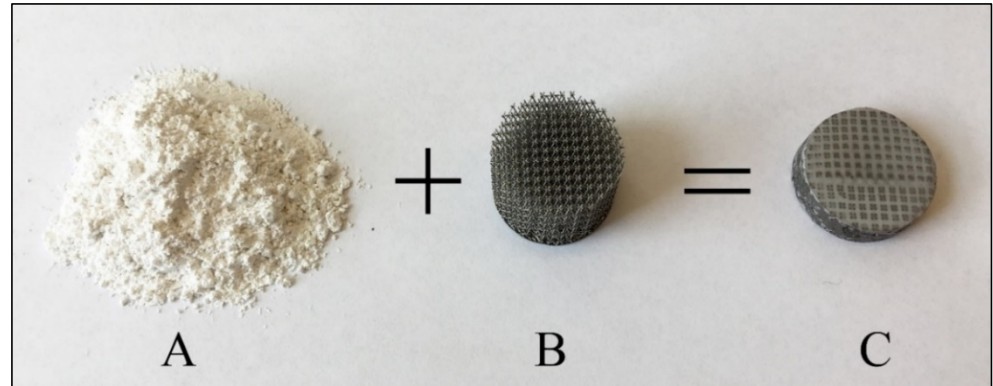

**Figure 8.** (**A**) Wollastonite (CaSiO$_3$) powder, (**B**) selective laser melting (SLM)-produced Ti6Al4V lattice structure with 1 mm cell size, and (**C**) Sintered sample (dimensions of lattice structure: 20 mm diameter and 15 mm height before spark plasma sintering (SPS) and 6 mm height after SPS and polishing).

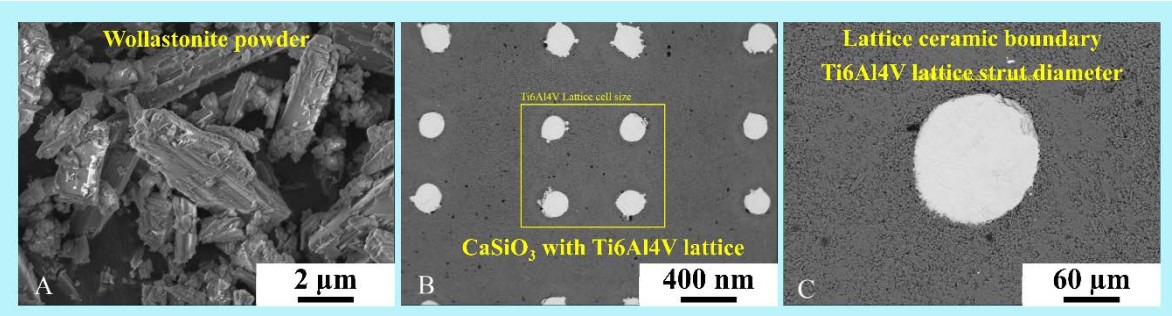

**Figure 9.** SEM micrographs of (**A**) wollastonite (CaSiO3) powder, (**B**) Sintered CaSiO$_3$ embedded in a Ti6Al4V lattice after SPS, and (**C**) High magnification image of metal-ceramic boundary.

## 4. Results and Discussion

An in vitro simulation and experimental results are required before an in vivo assessment of any biomaterial. From the literature, the ultimate strength of Ti6Al4V manufactured by SLM or electron beam melting (EBM) can be found between ≈1.0–1.2 GPa [34,35], and this value is applied for simulations in the current research. The rectangular scaffold (Figure 3) exhibits a high compressive strength, and deformation mostly occurs in the top and/or bottom part of the sample. The final height of rectangular and honeycomb after the compression test are similar. Comparison between Figures 3 and 7 present the findings that due to interlayer horizontal sheets, embedded multi-tubes (inspired by multi-walled carbon nanotubes) and hexagonal pores, the honeycomb scaffold has less stress concentration values; it is wrinkled with an expansion in diameter.

As expected, primitive, lattice and gyroid scaffolds (Figures 4–6) exhibited uniform deformation. For these three conditions, adding layers upon layers with a defined unit cell size (for example a lattice with intersecting cellular rods) produce desired structure shapes easily, but will be deformed relatively easily during compressive loading. It shows that cell type, size and alignment together play a pivotal role on their failure mechanism [36]. The experimental stress-strain curves from the compression test are shown in Figure 10. As seen from Figure 10, primitive, lattice and gyroid scaffolds show a more inferior deformation stress than in the case of rectangular scaffolds. In addition, the rectangular

scaffold shows uniform plastic deformation allowances, where the stresses were distributed uniformly and the local overloading is completely avoided. It is shown that rectangular (due to vertical channels and fluid permeability) and honeycomb (due to horizontal plates and hexagon pores) are structures can bear high compressive loads. As demonstrated in section views of Figure 11, rectangular scaffolds have vertical thin plates that are resistant against distortion and deformation, whereas, honeycomb is more suitable for alive cell growth due to horizontal plates and continuous open porosity.

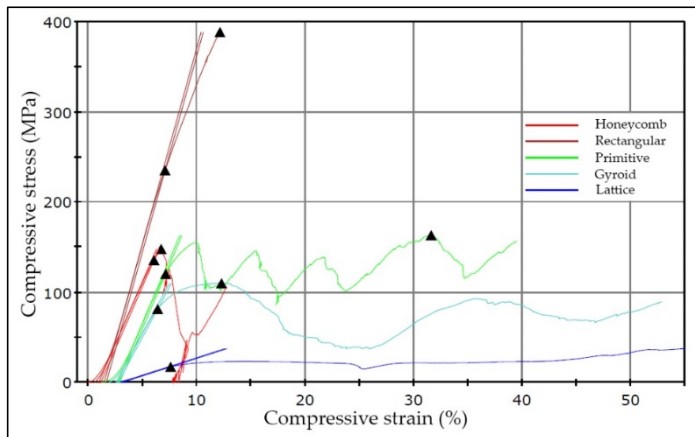

**Figure 10.** Stress-Strain compression results of scaffolds produced using the selective laser melting.

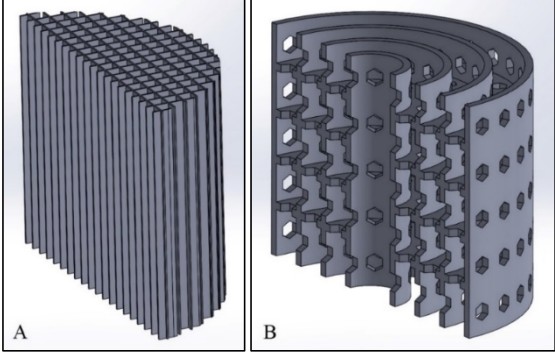

**Figure 11.** Section view of: (**A**) Rectangular, and (**B**) Honeycomb scaffolds.

The applicability of the porous TPMS scaffolds can be extended to applications related to (1) oxygen transport and/or (2) scaffold permeability [37]. The porous TPMS scaffolds can also contribute to enhanced cell seeding, and at the same time can maintain nutrient transport throughout the whole scaffold during in vitro culturing [16]. Regarding the final height of the deformed samples, the experimental compressive results (Figure 12A–E) are compared with the FEA analytical outcomes (Figures 3B, 4B, 5B, 6B and 7B), which are in good agreement. Regarding the Von Mises stress distribution (Figures 3A, 4A, 5A, 6A and 7A), primitive, lattice and gyroid scaffolds are relatively flexible/deformable structures, while more deformation energy can be absorbed by rectangular and honeycomb structures. Hence, for the rectangular structure, stress concentration and distortion can appear in the upper part of sample, but honeycomb failure started from the bottom of structures. Adding metallic lattice structure (Ti6Al4V) to ceramic reinforcement like wollastonite ($CaSiO_3$) produces a composite with higher wear resistance, damage tolerance and mechanical properties resulting in a higher durability of bones [38,39]. The strut diameter of the lattice is currently $\approx 200$ μm, while it could be increased up to 1 mm (Figure 9C). This ability helps to control volume fraction of both Ti6Al4V and $CaSiO_3$ in the composite required to achieve the density and porosity characteristic for bones. Besides, our SLM-SPS combination provides the possibility of making complicated shapes/structures for antibacterial or biomedical applications (Figure 8C). Height of scaffolds after the analytical and

experimental compression test are presented in Table 1. Statistical comparison shows a satisfying agreement between them. For Lattice scaffold (1 mm cell size and 0.5 mm strut diameter), the simulation outcomes show relatively lower weight than the experimental results. Such differences between the simulation and experimental results may be attributed to the porosity and unmelted/attached Ti6Al4V particles in the SLM parts [38,40].

**Table 1.** Compression test data (comparison of finite element analysis (FEA) simulation and experimental results).

| Scaffold Type | Weight Before Test (g) (Figure 2) | Volume Fraction (% of Metal) | Height after Simulation (mm) (Figures 3–7) | Height after Experiment (mm) (Figure 8) | Maximum Compressive Load (kN) (Figure 9) |
|---|---|---|---|---|---|
| Rectangular | 4.0 ± 0.1 | 19 ± 1 | 11 ± 1 | 11 ± 2 | 100 ± 5 |
| Primitive | 5.4 ± 0.2 | 28 ± 1 | 8 ± 2 | 8 ± 2 | 52 ± 2 |
| Lattice | 3.8 ± 0.1 | 17 ± 2 | 6 ± 2 | 8 ± 2 | 12 ± 1 |
| Gyroid | 4.0 ± 0.1 | 26 ± 2 | 7 ± 2 | 8 ± 1 | 35 ± 2 |
| Honeycomb | 5.6 ± 0.1 | 19 ± 1 | 11 ± 1 | 12 ± 3 | 70 ± 4 |

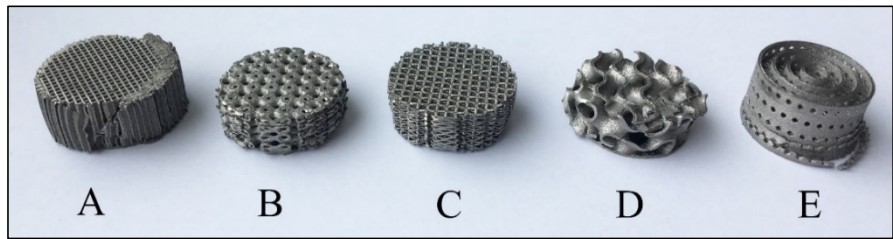

**Figure 12.** Triply periodic minimal surfaces (TPMS) scaffolds after compressive test with indication of their final height: (**A**) Rectangular −11 mm, (**B**) Primitive −8.4 mm, (**C**) Lattice −8.8 mm, (**D**) Gyroid −7.6 mm and (**E**) Honeycomb −12.1 mm.

Primitive TPMS have the highest volume fraction (28.2%, Table 1), and it might be interesting for biological activities due to a larger specific surface area. Stress distribution was uniform (Figure 4A), and shows a least difference between the simulation and experiment result (Figure 12B). Gyroid shows similar behavior as primitive, but including some broken/separated pieces in the top area because of a curvy cross section surface under compression. Strength of rectangular scaffold (Figure 12A) subjected to compressive loading is presented in Figure 9 and Table 1.

It survived under the maximum test load (100 kN). Comparison between A and Figure 12A shows that middle rectangular sections are less affected during compression, and any major deformation starts from the outer layers. Rectangular structure with rectangular/vertical channels is supposed for cells viability, oxygen transport and fluid permeability, otherwise it can be reinforced by horizontal plates (similar to honeycomb, Figure 11) for higher resistance against shear stress.

## 5. Conclusions

In this paper, five different Ti6Al4V triply periodic minimal surface structures with different surface areas were created by CAD design, namely rectangular, primitive, lattice, gyroid and honeycomb. The finite element simulation in comparison with 3D additive manufactured experimental results illustrated similar mechanical behaviors when the samples were subjected to compressive loading. They had uniform stress distributions and relatively identical displacements. It was found that ANSYS simulation has a potential to predict the mechanical behavior of additively manufactured scaffolds. Rectangular and honeycomb were novel cellular scaffolds designed for high compressive load-bearing and biological application with vertical and horizontal available surfaces, respectively. Rectangular scaffold is identified as suitable for oxygen transport and fluid permeability, whereas, honeycomb is

found to be the best for the growth of cells. SLM-manufactured Ti6Al4V lattice can be sintered via SPS along with $CaSiO_3$ for load bearing bone replacements.

**Author Contributions:** Conceptualization, R.R. and K.G.P.; methodology, M.A. and L.K.; investigation, R.R. and Y.H.; writing—original draft preparation, R.R.; writing—review and editing, M.A., L.K. and K.G.P.; supervision, M.A. and L.K.; funding acquisition, M.A., L.K. and K.G.P.

**Funding:** This research was supported by the Estonian Ministry of Education and Research under projects IUT19-29, the European Regional Fund, project number 2014-2020.4.01.16-0183 (Smart Industry Centre), ETAG18012, MOBERC15 and by base finance project B56 and SS427 of Tallinn University of Technology.

**Conflicts of Interest:** The authors declare no conflict of interest.

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
