# Peer review of "Mechanical Behavior of Ti6Al4V Scaffolds Filled with CaSiO3 for Implant Applications"

_applsci, doi:10.3390/app9183844_

Round 1

Reviewer 1 Report

This work investigates the mechanical behavior of Ti6Al4V scaffolds filled with CaSiO3, with five different model design (i.e. rectangular, primitive, lattice, gyroid, and honeycomb) by finite element simulation and 3D additive manufactured experimental results. This work is very interesting; however, minor revision is required to further improve the quality of the manuscript by addressing the following concerns.

The manuscript should be re-organized. Introduction should not contain figures; many figures in section 2 should be in section 3. Introduction: the progress of mechanical behavior of porous titanium structures should be described, such as Advanced Engineering Materials 20 (2018) 1700842; Acta Materialia 150 (2018) 1-15; Scripta Materialia 153 (2018) 99-103; Journal of Materials Science & Technology 32 (2016) 505-508; Acta Materialia 113 (2016) 56-67. Figures 4-7 do not give much values to the work. Table 1: all the experimental results should be with error bars. Discussion is desirable.

Author Response

We thank the reviewer for this positive criticism.

As suggested by the reviewer, the manuscript has been re-organized. The figure from the introduction has been moved to results and discussion. In addition, the figures in section 2 have also been transferred where and when necessary. Error bars were introduced in Table 1 and discussion is introduced. Hope revised version of the manuscript may now be accepted for publication.

Reviewer 2 Report

The presented paper investigates an interesting topic of composite structures from Ti64/CaSiO3 material obtained by a combination of SLM and SPS methods.

There are a few comments I'd like the authors to address:

In my opinion, Fig. 1–2 would suit better in the Materials and Methods section, while Fig. 3–7 would suit better in the Results section, where the FEA results are discussed. The Materials and Methods section does not have information regarding specimen preparation methods, the parameters used (for SLM, SPS, and characterization methods) and the starting materials. This information is provided further in the text, while it would fir better in the second section. What was the deformation of metallic structures after the SPS process? Is it possible to keep the intended design of such structures or achieve a desired pore size and distribution? One of the key properties for metallic scaffolds if Elastic modulus, especially, in case of Ti6Al4V alloy when its Elastic modulus is much higher compared to bone material. Can the authors provide any results regarding Elastic modulus of the obtained composite stuctures?

Author Response

As suggested by the reviewer, the manuscript has been re-organized. Fig. 1-2 is moved to the Materials and Methods section. Fig. 3-7 were shifted to section 3. All the necessary information about SLM, SPS and characterization methods were included in the experimental section. After the SPS process, the cell size may not be maintained, since pressure is applied to obtain a pore-free part. This is a competition between the deformation of the structure and the amount of porosity left in the sample. Hence, the aim here is to produce a defect free part (porosity free) and hence not much importance is given to the deformation of the structures during SPS process. We were not able to measure the elastic modulus of the parts, since some of them are very brittle and is difficult to carry out tensile properties to measure the modulus. Ultrasound based methods may be utilized, again the combination of metal-ceramic structure makes it very difficult and hence is not measured and included in the present manuscript. Hope the revised version of the manuscript may not be accepted for publication in the present state.